# Panel Products Made of Oil Palm Trunk: A Review of Potency, Environmental Aspect, and Comparison with Wood-Based Composites

**DOI:** 10.3390/polym14091758

**Published:** 2022-04-26

**Authors:** Arif Nuryawan, Jajang Sutiawan, Nanang Masruchin, Pavlo Bekhta

**Affiliations:** 1Department of Forest Products Technology, Faculty of Forestry, Universitas Sumatera Utara, Medan 20155, North Sumatra, Indonesia; jajangsutiawan@usu.ac.id; 2Department of Forest Management, Faculty of Forestry, Universitas Sumatera Utara, Medan 20155, North Sumatra, Indonesia; rahmawaty@usu.ac.id; 3Research Center for Biomass and Bioproducts, National Research and Innovation Agency (BRIN), Jl. Raya Bogor Km. 46, Cibinong 16911, West Java, Indonesia; nana021@brin.go.id; 4Research Collaboration Center for Biomass and Biorefinery, National Research and Innovation Agency (BRIN), Jl. Raya Bogor Km. 46, Cibinong 16911, West Java, Indonesia; 5Department of Wood-Based Composites, Cellulose and Paper, Ukrainian National Forestry University, 79057 Lviv, Ukraine; bekhta@nltu.edu.ua

**Keywords:** composite panel, biomass, oil palm trunk (OPT)

## Abstract

Oil palm plantations have expanded rapidly in Southeast Asia, particularly in Indonesia and Malaysia. A lot of products, including food and other edible products, oleo-chemicals, cosmetics, personal and household care, pharmaceutical products, and biodiesels are derived from palm oil, thus making them one of the most economically important plants. After 25–30 years of age, the palms are felled and replaced due to declining oil production. Oil palm trunks (OPT) are considered significant waste products. The trunks remain on the plantation site for nutrient recycling or burning. This increases insect and fungi populations causing environmental problems for the new palm generation or air pollution due to the fire. Up till now, OPT has received less attention in research studies. Therefore, this review summarizes the utilization of OPT into products made of oil palm fibers mainly derived from OPT and its application as the substitution of wood panel products. Some research works have been carried out on oil palm fibers that are derived from OPT for exploiting their potential as raw material of composite panel products, which is the objective of this review. Areas of development are processed into various conventional composite panel products such as plywood and laminated board which are usually predominantly made of wood and bonded by synthetic resins, particleboard with binder, or binderless and cement board which is arranged with wood as a minor component. All of the products have been presented and described technically according to best knowledge of the authors and literature review.

## 1. Introduction

*Elaeis guineensis* Jacq. is the broadest species in oil palm plantations for palm oil production. According to specimens that were collected in Martinique Island, it was native from the Guinea coast of West Africa and was formally named by Jacquin in 1763. The first commercial crops in estates were established in Sumatra Island-Indonesia and Peninsular Malaysia in 1911 [1]. Thus, this foundation made oil palm an important integrated industry, with both upstream and downstream sectors. Plantation as an upstream agriculture sector provides fresh fruit bunches (FFB) as raw materials for further processing in downstream mills such as vegetable oils and fat industries. They manufacture end-products for food and other edible uses (cooking oil, frying fats, margarine, confectionery/cocoa butter substitute, ice cream, non-dairy creamer, salad oil, shortening, etc.) [2]; oleo-chemical (stearine, fatty acids, fatty alcohols, fatty methyl esters, and glycerin) [3]; cosmetics, personal care, and household cleaner (balms, lipsticks, emulsifiers, soaps, detergents, candles and cleaning products) [4]; skin-based pharmaceutical products (vitamin E supplement, tocopherol, and tocotrienol) [5]; and biodiesel production (lubricant, surfactant, transesterification of triglycerides with alcohol) [6,7].

Since palm oil is the primary yield of oil palm plantations and economically considerable revenue is gained through this commodity, these crops have expanded dramatically. Nowadays, worldwide 43 countries have cultivated this palm [8], and almost 80% of the world oil palm plantation is centered in Southeast Asia [9], with most of it occurring in Indonesia, Malaysia, and Thailand [10]. Both Indonesia and Malaysia are the two major palm oil-producing countries and the two top world producers of oil palm products, having the largest planted oil palm area in the world [11]. Figure 1 illustrates their development in the last decade. The data area harvested and oil palm production in the predominant country in Southeast Asia and the world a decade back showed an increase [12].

Both Indonesia and Malaysia produce 85% of the world’s palm oil [13], where oil palm estates occupy 22 of 34 provinces in Indonesia [10] and cover more than 60% of the agricultural land of both the west (the peninsular) and the east (the States of Sabah and Sarawak) in Malaysia [13]. The large extent of oil palm plantations in both countries has generated waste products in large quantities. For instance, the fibrous part can be gained after the removal of oil seeds from FFB for oil extraction which results in crude palm oil (CPO) and crude palm kernel oil (CPKO) that are derived from the mesocarp fiber (MF) and the palm kernel (the seed enclosed in a shell of the endocarp), respectively (Figure 2) [13].

In addition to oil palm processing, various waste consisting of empty fruit bunches (EFB), palm kernel shells (PKS), and mill effluents (ME) can also be obtained [9]; however, these are not under the scope of the present study and thus will not be discussed here because all of these only credited 10% of biomass [11] and are derived from by-products of the industrial process. This study discusses a large amount of biomass that is produced from oil palm growing on plantations, either during pruning or replanting as shown in Figure 3. 

Biomass refers to any organic plant products with general uses [10] categorized as agro-based resources [11]. They have broad characteristics such as being renewable, widely distributed but locally available, voluminous or bulky, moldable, variety in many forms and sizes, anisotropic, hygroscopic, recyclable, versatile, non-abrasive, porous, biodegradable, combustible, compostable, and reactive. They also have a high aspects ratio, high strength to weight ratio, relatively low energy conversion, and good insulation properties (sounds, electrical, and thermal). The general fiber structure is hollow, laminated with molecular layers, and an integrated matrix [14,15,16].

Even though there are two primary biomass residues, namely oil palm frond (OPF) with oil palm leaves (OPL) and oil palm trunk (OPT) in oil palm growing on plantations [17], only OPT was considered in this review. OPF with the leaves which are generated during harvesting of FFB has been used as a substitute for grass as well as the source of roughage for decades as for ruminant feed such as for goats, beef, and dairy cattle without any report of toxicity [18] where the farm sites are usually located near the oil palm plantations. Long-term feeding of OPF-based feeds, either fresh or processed as silage or pellet, has produced good quality carcasses, and the meat is safe for consumption [19]. However, OPT has received less attention in research studies [20] even though the potency is abundantly available. OPT is available only when the economic lifespan of the palm is achieved at the time of replanting. The average age of replanting is approximately 25 years when OPT is discarded for re-plantation after 25–30 years of oil production [8]. The main economic criteria for felling are the height of the palm, reaching 13 m or above years, either due to their decreasing yield or because they have grown too tall, which makes harvesting very difficult [21]. During replanting, the diameter of the felled trunk is around 45 cm to 65 cm, measured at breast height [9]. In other words, the OPT is fallen during replanting when the stems have reached the end of their useful life [22]. It is a less expensive lignocelluloses raw material as compared to wood. Using OPT as a raw material to produce value-added products will reduce the overall costs of production and increase economic returns [9]. Further, OPT is a sustainable non-wood source that is available in abundance at replanting operations would have seemed for alternative raw materials for some of the products that are traditionally made from wood such as veneer, sawn lumber, and fiber-based bio-composites with considering position on the trunk [23]. Unfortunately, OPT is monocotyledon and consists of vascular bundles and parenchyma, which are the main obstacle in utilizing OPT as a wood solid. In monocotyledon wood, fibers have less strength and are irregular in characteristics compared to the dicotyledonous wood. OPT wood consists of primary tissue, and it is not comparable to dicotyledons and gymnosperms in terms of wood development because oil palm does not possess cambium. Indeed, OPT is not wood compared to softwood and hardwood, which wood-based industries are used to [24]. Khalil et al. [24] mentioned that fibers within monocotyledons have had less strength and irregular properties compared to wood in dicotyledons. Therefore, in this regard, OPT would be utilized as alternative wood products because so far, many studies of OPT could be engineered into the substitution of traditional wood products such as plywood, laminated veneer lumber (LVL), particleboard, fiberboard, and cement board predominant matrix which will be discussed later.

In this review, the potential for transforming the waste biomass of OPT to an alternative substitution of conventional wood products was discussed, including their environmental aspects. Based on the availability of OPT and the environmental and economic impacts of the fibrous parts from OPT, this study’s objective is to evaluate the quality of composite panel products that are made of OPT to be an alternative substitution of wood products. The success of these alternative products will be supported by previously published studies, performed prototypes, and standard specifications.

## 2. Method

A review of the literature found 599 titles on sciencedirect.com (accessed on 21 December 2020) and 74 titles via a search engine in the target journal Polymers (MDPI). After reviewing the title, abstract, and content of the article to ensure that the raw materials that were used to create the composite products that were derived from OPT were listed, there were only 31 titles on the search engine sciencedirect.com and 2 titles on the journal Target Polymers (MDPI), as presented in Table 1.

From this literature review, OPT or OPS has been used as the main raw material of various composite panel products ranged from tiny size such as fine particles and fibers up to small dimensions such as chips. Moderate size was found as strands of OPT, resulted in a separation between vascular bundles and the parenchyma. Larger dimensions were found in the form of veneer mats and veneer lumber. These materials are used to make composite panel products or bio-composites, which often replace wood-based panel goods such as particleboard, fiberboard, chipboard, oriented strand board (OSB), plywood, and LVL. Sometimes, OPT was found as minor part, such as filler or substitution of aggregate in cement board, gypsum board, or concrete brick. The type of binder that was used ranged from binderless (without adhesive), using bacteria (PHA), thermoplastics (PVA, PVAc), up to conventional formaldehyde-based resins (UF, MF, PF). In mineral boards, cement or gypsum has a role as both a binder and a matrix. Discussions further about these raw materials will be conveyed in next sub chapter.

## 3. Environmental Aspect

Currently, Malaysia’s oil palm industry generates more than 80 million tons of oil palm waste per year (dry weight basis) [58,59], with the figure expected to increase by at least 40% by 2020 [60]. Annual oil palm production is expected to increase by 50 million tons by 2030 in response to Malaysia’s proposed expanding oil palm plantations [61]. The most frequently generated co-products that are derived from fruits include palm kernel shell (PKS) [62], OPEFB, mesocarp fiber (MF), and palm oil mill effluent (POME) while from the non-fruit include oil palm frond (OPF), oil palm leaves (OPL), and OPT [63,64,65]. Table 2 shows the comparison of oil palm biomass residues that originated from pruning, replanting, and milling activities between the two highest producing countries of oil palm, namely Indonesia [22] and Malaysia [66].

Malaysia is internationally renowned for its palm cultivation, and it is a runner up in palm oil production and exports [67] after first rank of Indonesia. It accounts for more than 30% of the global palm oil production and 37% of global exports in 2016 [68]. Apart from that, the oil palm industry and palm oil mills generate approximately 90% of biowaste, even though the oil accounts for only 10% of the biomass [69]. As a result, large quantities of oil palm biomass are generated each year and are then discarded or burned without further use due to a lack of technology to utilize this material [70]. This situation exacerbates the problem of biomass overload, wastes valuable cellulose-rich resources, and may result in serious environmental problems such as air pollution [71]. Additionally, the improper handling of this residue and biowaste may raise environmental and health concerns. It may contribute to eutrophication, pollution, and other types of disturbances for aquatic and terrestrial life [72].

The biomass data that originated from the extraction of fruit (FFB) was dependent on palm oil mills, their operation, and capacity. In Indonesia there were 421 mills [22] spread out in 18 provinces in five big islands (Sumatra, Java, Kalimantan, Sulawesi, and Papua) while in Malaysia it was slightly higher about 454 mills [66] that were concentrated in Peninsular Malaysia, Sabah, and Sarawak. The data of OPF was predominant in both countries because the fronds (and leaves) were harvested during the regular pruning through the lifetime of the palm. OPT or OPS were collected after the palm exceeded its economical age about 25–30 years. The traditional method for establishing new oil palm plantations or replanting is through mechanical or fire clearing of the land. Mechanical clearing is frequently required in large-scale plantations, resulting in soil compaction and other physical degradation of the soil. When old palm stands are felled, stacked, and burned, a large amount of smoke is released into the environment. Slashing and burning removes the aboveground biomass, understory vegetation, and ground litter, resulting in high environmental costs. The purpose of burning biomass in the land clearing process is to eliminate waste material that obstructs plantation management and eradicate pests and diseases by destroying their breeding medium.

Additionally, burning of the unwanted biomass and another waste material appears to be the most cost-effective and time-efficient method of waste disposal. This activity results in a large amount of CO_2_ being released into the atmosphere, contributing to climate change and global warming. Particles that are produced by the burning undoubtedly contribute to the haze problem. Fortunately, all these wastes are classified as organic wastes that degrade naturally. However, because these wastes are generated in large quantities, they can pollute the environment.

The presence of this waste that is not managed correctly promotes the growth of Ganoderma disease. Vascular wilt, bud and spear rot, sudden wilt, red ring disease, and basal stem rot disease are the most common diseases that attack oil palm trees. The basal stem rot (BSR) disease poses the greatest threat to oil palm plantations. BSR disease, also known as Ganoderma disease, is caused by a pathogenic basidiomycete, a white-rot fungus called *Ganoderma boninense*. Indonesia and Malaysia have a high prevalence of BSR, whereas Africa, Papua New Guinea, and Thailand have a low prevalence [73]. According to a Malaysian study, oil palm stands that are planted after coconut trees in coastal areas are more susceptible to this disease. According to reports, up to 50% of yields have been lost, and 80% of oil palm has been killed before their economic age [74]. In comparison, another survey discovered that incidences of BSR infection in Malaysia, Peninsular Malaysia, Peninsular Malaysian Coast, Sarawak, and Sabah were 29, 34, 37, 9%, and 29%, respectively [75]. In Indonesia, a survey that was conducted in an estate in North Sumatra found that 22% or about 30 palms per hectare died after being infected this disease causing a significant loss in yield production as well as threatening palm oil sustainability [76].

Ganoderma disease begins with the pathogenic fungi colonizing the oil palm root, followed by the degradation of the basal stem tissue [77]. The disease then destroys the internal tissue and the palm xylem, disrupting the water and nutrient supply from the root to the plant’s upper parts [78]. Infected stands will show signs of wilting, desiccated fronds, and unopened spears. Later, basidiocarps or fruiting bodies in the shape of a conch will develop on the palm trunk. At this point, the fungal infections have spread widely throughout the palm, typically resulting in the plant’s death [79]. The fruiting body can then release spores and spread to the soil or other palms. As a result, Ganoderma reduces the useful life of oil palms and results in significant yield losses for the oil palm industry [80]. The Malaysian yield losses due to this disease are estimated to be up to USD 500 million, with an average mortality rate of 3.7% [81].

If no effort is made to utilize oil palm biomass, it will become a problem for the environment. When it is used correctly, it will eliminate disposal issues and generate value-added products. As a result, the oil palm industry must be prepared to capitalize on this situation and maximize the utilization of available oil palm biomass to convert waste to wealth [82]. This biomass residue may be used in place of wood to manufacture wood-based panels [83]. Worldwide, research has been conducted to maximize the economic value of biomass residues through the production of lumber, furniture, and lightweight construction. Malaysia pioneered oil palm biomass to create medium-density fiberboard (MDF) [84]. Globally, an increased awareness of oil palm biomass-based products has increased their demand [85].

## 4. Overview Production of Wood-Based Panels in the World

Global wood-based panel production totaled 408 million m^3^ in 2018, a 1% increase over the previous year (404 million m^3^) and a 9% increase over the previous nine years. Wood-based panels were the fastest growing product category in production, owing to the Asia-Pacific region’s rapid and consistent growth up to 2016. Global production has stabilized in recent years. In 2018, Asia-Pacific accounted for 61% of the global output (248 million m^3^), followed by Europe (90 million m^3^, or 22%), Northern America (48 million m^3^, or 12%), Latin America and the Caribbean (19 million m^3^, or 4%), and Africa (3 million m^3^, or 1%) [86].

Since 2014, the global trade in wood-based panels has been steadily increasing. In 2018, it increased by 3% to 91 million m^3^, accounting for 22% of the total production. In 2018, Europe and Asia-Pacific dominated the international trade in wood-based panels, accounting for 71% of total imports and 80% of total exports, respectively. In 2018, the top five producers of wood-based panels (China, the United States of America, Russia, Germany, and Canada) accounted for 69% (282 million m^3^) of global production. Meanwhile, the top four consumers of wood-based panels are the same as the top four producers, indicating that the products are primarily consumed domestically. Consumption trends are parallel with the production trends [86].

Plywood (which includes blockboard and LVL) produced 163 million m^3^ (or 40% of the total wood-based panel production) in 2018, an increase of 11% over 2014. Regional variations in the composition of various wood-based panel products exist. In North America and Europe, reconstituted panels (OSB, particleboard, and fiberboard) dominate other product categories, whereas plywood (including blockboard) is the most widely used wood-based panel product in Asia-Pacific (mainly in China). Each primary wood-based panel product accounts for roughly the same proportion of the total production in Latin America and the Caribbean as the others [86].

Between 2014 and 2018, OSB and particleboard production increased at the fastest rates, by 25% and 13%, respectively. Both products multiplied in Eastern Europe, most notably in the Russian Federation. The global fiberboard production peaked in 2016 (121 million m^3^) and then fell 4% to 116 million m^3^ in 2018, remaining at 2014 levels. Between 2014 and 2018, both MDF and high-density fiberboard production increased by 5%, accounting for 85% of total fiberboard production. Other types of fiberboards saw a 19% decline during the same time [86].

The development of wood-based panel production in Indonesia and Malaysia is shown in Figure 4. Wood chip and particle production in Indonesia and Malaysia have experienced the same characteristics, which have increased in the last decade (2009–2019). However, there are differences in veneer sheet and plywood production in Indonesia and Malaysia. The production of veneer sheet and plywood in Indonesia has increased, while veneer sheet and plywood in Malaysia has decreased in the last decade. These phenomena are due to changes in industrial development from plywood to particleboard in Malaysia, which has increased. Meanwhile, the development of OSB in the two countries has not been done much. Therefore, this is very interesting to study the development of the OSB industry in Indonesia and Malaysia.

## 5. Types of Wood-Based Panels

### 5.1. Particleboard

Particleboard is a broad term that refers to any panel products that are made entirely of wood particles. Naturally, particleboards are composed of a diverse range of particle shapes and sizes. As a result, the particle type is used to define the type of particleboard product. For instance, particleboard is a generic name for products made of particle while particular size of chips, wafer, or strands resulted in chipboards, waferboards, or OSB, respectively. Another characteristic of particleboards is that the wood particles are bonded together using a synthetic resin adhesive and then compressed at high pressures and temperatures. This is critical because the manufacturing process has a significant impact on the properties of the resulting panel products [87].

### 5.2. OSB

OSB are multilayered panels that are made from pre-shaped strands of wood (typically 15–25 mm wide, 75–150 mm long, and 0.3–0.7 mm thick) that are bonded together under pressure and heat with a binder (often water-resistant). The strands of the outer layers run parallel to the long board edge and the production line. While the strands in the core layer are frequently smaller and can be oriented randomly or at right angles to the strands in the face layers, the strands in the core layer are frequently larger and can be aligned at right angles to the strands in the face layers. OSB is classified into four categories by the European Norms (EN) 300 and EN 13986 standards: OSB-1 is used for general indoor (dry) purposes, OSB-2 is used for load-bearing purposes in dry conditions, OSB-3 is used for load-bearing purposes in humid conditions, and OSB-4 is used for heavy-duty construction in humid conditions. OSB is typically available in thicknesses ranging from 10 to 32 mm. The internal bonding (IB) and modulus of rupture (MOR) tests following cyclic boiling are frequently the most difficult to pass for OSB-3 and 4, which is why manufacturers use moisture-resistant resins such as isocyanates (PMDI), phenolic-based resins (PF, MUPF), or UF resins that are reinforced with melamine (MUF). MOR and modulus of elasticity (MOE) values that are parallel to the long panel edge are typically double those that are observed across the panel. This is due to the strand orientation of the face layer [87].

### 5.3. Fiberboard

MDF is an engineered wood product that is composed of fine lignocellulosic fibers with or without synthetic resin to form panels. Binderless fiberboard utilizes a wet manufacturing process for activating hydrogen bonding within the panels. While wood is the most common lingo-cellulosic fiber, other plant fibers such as bagasse and cereal straws can also be used [87]. MDF is available in a variety of panel sizes and thicknesses ranging from 2 to 100 mm. These panels have a density of between 500 and 900 kg/m^3^. The faces of the panels are smooth and dense (for dry manufacturing process using synthetic resin), and the color ranges from pink-brown to dark brown unless a dye is added during manufacture [87].

### 5.4. Plywood

This product is a flat panel that is constructed from veneer layers of either soft or hardwood, which are arranged perpendicularly in odd numbers and then glued either using traditional thermosetting resin or using thermoplastics. It is stacked together to form a panel using a clamp or a pressing machine. Each layer is given a unique name; for example, the outer layer is referred to as the face, the inner layer as the middle or core, and the wood grain layer that is perpendicular to both the face and back sides is referred to as the crosswise [88]. The properties of plywood are determined by the quality of the veneer layers, the layering order, the adhesive that is used, and the control of bonding conditions.

## 6. Comparison Wood versus Oil-Palm Trunk Biomass

### 6.1. Physical Properties of Oil Palm Trunk

According to Srivaro et al. [89], the water uptake (WU) of the OPT ranges between 73 and 415%. According to Srivaro et al. [90], the WU increases gradually along with the trunk height and the central region. The outer and lower zones have significantly lower values than the other two zones. The increasing trend in WU can be attributed to the distribution of parenchymatous cells, which retain more moisture than the vascular bundles. According to Lamaming et al. [91], between the parenchyma tissues and vascular bundles are comprised of different compositions of extractive, lignin, and holocellulose. By contrast, parenchyma tissues are more abundant near the apex of the OPT and the central region. Recently, Wulandari and Erwinsyah [92] reported that the highest moisture content (MC) was found at the inner and gradually decreased toward centre to the trunk’s outer part. These percentages ranged from 36 to 141%. A decrease in the percentages of parenchyma cells with a high capacity for water absorption (WA) increased the number of vascular bundles. Generally, Wulandari and Erwinsyah [92] stated that the MC of the OPT decreased as the trunk height increased.

Due to the oil palm’s monocotyledonous nature, its density values vary significantly across its trunk. Density, or specific gravity, is regarded as the most critical characteristic when utilizing OPT. It is defined as the quantity of material that is contained in a unit volume. Generally, the density is the highest at the bottom end’s peripheral region and lowest at the top end’s central core region. The density values range from 222 to 404 kg per m^3^ [90]. According to Srivaro et al. [90], the density of an OPT decreases linearly with trunk height and increases toward the trunk’s center; the outer region of the trunk has a density that is more than twice that of the inner region. A variety of factors cause these variations. For instance, the density across the trunk is primarily determined by the number of vascular bundles per square unit, which decreases towards the center. However, there were variations in the density along the trunk height since the vascular bundles at the top of the palm are younger.

### 6.2. Chemical Properties of Oil Palm Trunk

The chemical composition of OPT varies longitudinally and radially along the stem including the barks. No discernible trend exists for holocellulose, lignin, and ash content in bottom, middle, and upper parts of the oil palm meristem. However, they were significantly different from those of its bark [93]. The alpha-cellulose content, ash content, pentosan, and holocellulose within the OPT were lowest compared to those of OPF and EFB. Extractives within OPT which were indicated by water solubility (either cold or hot) and alcohol- and alkali-soluble components were in between of those of EFB and OPF [94].

Further, Onuorah et al. [94] examined the chemical properties of OPT. They reported that OPT contains more lignin than those of EFB and OPF. Ahmad et al. [95] observed that the bottom portion contained a higher lignin concentration and was significantly different than middle and top parts. This is probably due to the increased number of fibrous vascular bundles in the peripheral region [92]. These vascular bundles contain thicker and older cells, contributing to OPT’s lower end’s higher lignin content. The range of lignin content is between 17.21% and 20.47%. The ash content, which is also consistent throughout the trunk, ranges between 2.18 and 2.86% [95].

Additionally, free sugars between 2–14% were observed throughout the stem. The inner regions have a higher proportion of free sugars whereas the outer zones have the lowest free sugar content. Sucrose, glucose, and fructose are the three major free sugars in the OPT, as determined by high-performance liquid chromatography (HPLC), the most used method for analyzing sugar types and concentrations. Sucrose, the primary sugar component, remains relatively constant throughout the stem’s length. On the other hand, the glucose levels decrease as the stem height increases, whereas the fructose levels increase. However, a high variation of sugar with different content that were found in different oil palms were influenced by different methods of analysis, quantifications on different locations of the palms (inner or outer part, height), including different storage and microbial infestation [96].

### 6.3. Mechanical Properties of Oil Palm Trunk

The MOE which measures the material’s stiffness or rigidity, varies between 740 and 7960 MPa in the OPT, with the highest value found in the lower peripheral region. The lowest MOE is found in the central region’s core, near the trunk’s top end. The MOR, which varies similarly to the MOE, is extremely low compared to conventional timber species. Srivaro et al. [89] examined the mechanical properties of a 25-year-old OPT and compared them to those of other species such as bamboo and hardwood. The mechanical properties of the OPT reflect the density variation that was observed in both the radial and vertical directions of the trunk. The peripheral lower portion of the trunk has the highest bending strength, while the central core of the top portion of the trunk has the lowest. OPT has had the lowest bending strength, therefore OPT is more likely to be recommended for non-structural purposes.

The variation in the MOR follows the same pattern as the variation in the MOE or bending strength. The value of each strength was lower compared to bamboo, they ranged from 53 to 275 and 7 to 58 MPa (MOR), and 5 to 22 and 0.5 to 7.0 GPa (MOE). OPT has had a lower density compared to bamboo. Wulandari and Erwinsyah [92] investigated the density and specific gravity of Dura x Pisifera species of oil palm. They reported that the average values at the peripheral, central, and inner were 0.73, 0.54, 0.48 g/cm^3^ and 0.53, 0.26, 0.19, respectively.

### 6.4. The Anatomy of Oil Palm Trunk

The oil palm plant is not a woody plant. The oil palm is a monocotyledonous species; it lacks cambium, secondary growth, growth rings, ray cells, sapwood, heartwood, branches, and knots. Oil palm has an anatomical structure that is composed primarily of fibers, tracheids, vessels parenchyma, and ray parenchyma cells. OPT differs chemically from hardwood/softwood species, with differences in cellulose, hemicellulose, and lignin content [89].

The increase in the stem diameter occurs due to overall cell division and enlargement in the parenchymatous ground tissues and enlargement of the vascular bundle fibers. There are three primary sections. The cortex is first, followed by the peripheral zone, and finally by the central zone in the OPT [96,97]. Oil palm stands are typically replanted after 25 years. At replanting age, the OPT reaches a height of 7 to 13 m and a diameter of 45 to 65 cm, measured 1.5 m above ground level. When mature, the trunk tapers toward the crowns, which typically produce about 41 fronds. The anatomical characteristics of an oil palm trunk cross-section are described as follows:1.Cortex
The cortex is a narrow zone measuring approximately 1.5–3.5 cm in width, covering most of the trunk. It is primarily composed of ground tissue parenchyma with numerous longitudinal fibrous strands and vascular bundles of small and irregular shape [90]. In order to better explain the two main parts within OPT, in Figure 5 exhibits the schematic separation between the two for optimization further.

2.Periphery

The periphery is a region with thin parenchymal layers and dense vascular bundles that provides the palm trunk’s primary mechanical support. The peripheral region is usually densely packed with radially extended fibrous sheaths, providing the palm with mechanical strength. This region accounts for approximately 20% of the total cross-section’s area. The secondary walls of the fibers are multilayered and increase in length from the periphery to the pith. As the basal portions of the stem are older, they typically have more developed secondary walls than the top portions. Between the xylem and fiber strands are phloem cells in a single strand. According to Choowang [97], the areal number density of vascular bundles in the outer zone ranged from 68 to 115, much more compared to in the middle which had only 44 to 58 vascular bundles population/cm^2^.

3.Central

The central zone accounts for approximately 80% of the total area and comprises of slightly larger and widely dispersed vascular bundles that are embedded in the parenchymatous ground tissues. Bundles become larger and more dispersed as they approach the trunk’s core. An estimation of the vascular bundle density in central region’s is approximately from 23 to 40 vascular bundles population/cm^2^ [97].

4.Vascular bundles

The vascular bundles are thick, fibrous, and have a low hygroscopicity. The vascular bundle, on the other hand, is significantly less tightly packed toward the center zone, which contains more storage tissue. Fibers, xylem, vessels, protoxylem, sieve tubes, axial parenchyma, stegmata or spherical-shaped silica bodies form, and companion cells are all found inside the vascular bundle [91,98].

5.Parenchymatous tissue

The parenchymatous ground tissue is predominantly made up of soft, spongy, and very hygroscopic of thin-walled spherical cells. Food is stored as carbohydrates in the live parenchyma cells, mostly in the form of sugars and starch. The ground parenchymatous tissue is made up of thin-walled spherical parenchyma cells that are dense and thicker in the core than in the periphery. The diameter and length of the oil palm trunk increases as a result of cell division and elongation in the parenchymatous ground tissues as well as vascular fiber [91].

*Picea abies* (Norway spruce), a softwood species, has a different anatomy than the oil palm plant. Since it is a softwood, it contains earlywood (EW) and latewood (LW), which are readily visually visible. The LW is denser than the EW, and when viewed under a microscope, the cells of dense latewood have very thick walls and tiny cell cavities, whereas those of EW have thin walls and large cell cavities [99]. Macroscopically, the spruce wood consists of sapwood and heartwood whereas microscopy works revealed that wood was also arranged by tracheids which are typically ordered in a regular pattern and are nearly perfectly rectangular in cross-section [100]. Further, in spruce wood, abnormality in growth such as formation of hazel wood could be occurred [101]. In OPT, there is no differences among the bottom, middle or upper parts as shown in the scanning electron microscopy (SEM) image of Figure 6.

Therefore, whole parts utilization of OPT is beneficial compared to wood. However, further treatment should be taken, for instance in separating between the vascular bundles and the parenchyma. In the next sub chapter, the properties of OPT for raw material of panel products are discussed.

## 7. Composite Panel Products Made of Oil-Palm Trunk and Its Properties

OPT has several disadvantages: low strength, low durability, poor dimensional stability, and poor machining [99]. Due to the high-density gradient along the stem’s radial and longitudinal axes, OPT is less desirable for use as lumber. As a monocotyledon, oil palm differs significantly structurally from dicotyledonous woods. In comparison to wood, oil palm is much more porous, cellular, and anisotropic. As a result, liquids such as water and low molecular weight compounds can be rapidly absorbed and transported into the cell wall and lumen. OPT has been developed for composite panel products such as traditional panels (plywood and laminated veneer lumber), particleboard, and cement board.

### 7.1. LVL and Plywood

The first development for a traditional composite panel from the OPT was carried out by Sulaiman et al. [40]. The purpose of this study was to determine the physical and mechanical properties of LVL that was manufactured from OPT bonded by UF, PF, melamine urea formaldehyde (MUF), and phenol resorcinol formaldehyde (PRF) adhesives. Their shear strength constructed with PRF adhesive is greater than that of panels that were bonded with alternative adhesives. The shear strength of oil palm LVL, on the other hand, was lower than that of rubberwood LVL. There are few studies on OPT plywood constructed with thermoplastic adhesives. Recently, Hashim et al. [102] reported that LVL that was manufactured from OPT performed better when adhered with a cold setting adhesive, specifically PVAc and emulsion polymeric isocyanate (EPI). PVAc and EPI, with and without toluene, were superior to rubberwood LVL in TS and WA.

To improve the poor properties of traditional composite panels that are made from OPT, the raw materials have been modified. Numerous modifications were made, including treatment with low molecular weight phenol-formaldehyde (LmwPF) resin, impregnation of OPT with UF, hot air, and microwave drying, hybridization with another material, modification of the adhesive with nanoparticles, and compression of binderless veneer panels using response surface methodology (Table 3).

The pre-treatment of OPS veneers with LmwPF resin enhanced the surface characteristics, density, termite, and decay resistance. Loh et al. [38,50] investigated the wettability, surface roughness, and protection of OPS veneer from subterranean termites (*Coptotermes curvignathus*) and white-rot fungi (*Pycnoporous sanguineus*). For more than 30 s, the phenolic-treated veneer surface was capable of preventing liquid penetration. Veneer that was treated with LmwPF resin increased the durability of OPS plywood by 38% and 62% against termites and white-rot fungi, respectively. However, outer-layer OPS plywood is more resistant to termites and fungal decay than inner-layer OPS plywood.

Another study successfully developed the veneer treatment with LmwPF resin by varying the resin treatment, pressing time, solid resin content, and the resin concentration on formaldehyde emission, physical, and mechanical properties of plywood [42,45,47]. The plywood that was treated with LmwPF resin outperforms commercial OPS plywood in terms of physical and mechanical properties. The results indicated that veneers that were treated with a 30% solution of LmwPF provided the optimal treatment combination for plywood’s physical and mechanical properties. The pressing time affected the pre-preg OPS plywood’s mechanical properties and bonding performance. When LmwPF resins with a solid content of 32% or greater were used, the bonding quality met the minimum requirement for interior and exterior applications that are specified in Standard EN. The LmwPF resin can be used to treat OPT veneer with a solid content of up to 32% that meets the F standard for wood products as defined by Japanese Agriculture Standard (JAS).

Another successful treatment method to enhance the plywood’s quality was used on oil palm veneer (OPV). Lekachaiworakul et al. [46] examined the effects of hot air and microwave drying on oil palm veneer’s kinetics and mechanical properties of OPV. Additionally, drying with hot air and a high-power microwave revealed increased tensile stress and shear properties. However, the mechanical properties of hot air at 70–90 °C when combined with a microwave power level of 2000 watts were superior to those of hot air at 70–90 °C when combined with a microwave power level of 3000 watts.

Hybridization has been shown to affect the bending strength in variable studies. By crossbreeding of OPEFB and OPT, plywood’s bending strength, screw withdrawal strength, and shear strength are increased [103]. Srivaro et al. [54] investigated the stiffness and bending strength of an OPT core sandwich panel that was overlaid with a rubberwood veneer when bent in the center. An increased OPT core density improved the stiffness and strength of the beam. Face fractures and core shear failures were observed, with the latter occurring more frequently when the OPT core density was low, the veneer face was relatively thick, and the span length was short.

Furthermore, H’ng et al. [36] examined the effects of UF impregnation on OPT’s physical and mechanical properties by varying the UF solid content impregnation pressure and time. The three-layered OPT board that was impregnated with a higher resin solid content and a shorter impregnation time eventually reduced the degree of WA, resulting in changes in thickness and width. Additionally, impregnation treatments on the core layer increased the MOR and shear strength of the board but decreased the MOE of the board.

Previously, researchers manufactured plywood using formaldehyde-based resins adhesives. The immediate impact of the life cycle assessment (LCA) was due to the use of UF and PF during the plywood manufacturing process [103]. As a result, formaldehyde causes health problems such as skin, eye, nose, and throat irritation, and excessive exposure may cause certain types of cancer [104]. Low formaldehyde emission of UF has been developed by our group [105]. However, application of this resin was not using OPT. Our group applied PF resin for bonding OPT and OPEFB to create a hybrid plywood with oil palm ash (OPA) nanoparticles and PF resin as a binder. The presence of OPA nanoparticles significantly altered the physical, mechanical, and thermal properties of the plywood panels. Thermal stability testing revealed that PF-filled OPA nanoparticles contributed to the plywood panels’ thermal stability. Thus, the results of this study demonstrated that OPA nanoparticles significantly improved the properties of hybrid plywood.

Another study is being conducted to determine how to reduce formaldehyde emissions. Saari et al. [26] used a response surface methodology to produce compressed veneer panels from OPT without synthetic adhesive. Morphological analysis revealed that the shape of the vascular or parenchymatous tissue was altered as it was cranked, collapsed, or compressed together during the manufacturing process. Specific manufacturing parameters, such as pressing temperature, pressing time, and a portion of the oil palm trunk veneer, were demonstrated to bind the oil palm trunk veneers without using synthetic adhesives with acceptable mechanical and physical properties.

### 7.2. Particleboard

Table 4 summarizes the various studies and modifications that are made to the OPT panel. Hashim et al. [59] successfully developed the world’s first particleboard that was made from OPT. Hashim et al. [35,37] investigated the properties of panels that were made entirely of OPT without the use of binders, focusing on the effect of particle geometry and press temperature. The particle’s size and shape affect the sample’s properties. According to the findings of this study, increasing the pressing temperature may be a viable method for improving the properties of binderless particleboard. Oil palm biomass contains significant amounts of holocellulose, lignin, starch, and sugar, that are necessary for self-bonding adhesion.

Additionally, Lamaming et al. [49] conducted a successful study which demonstrated superior mechanical and physical properties for binderless panels that were made from young oil palm trunk particles. Additionally, sugar was found to improve the boards’ overall properties. The age of the oil palm affected the mechanical and physical properties of the boards that were manufactured, with sugars playing a significant role in the boards’ self-bonding.

There are several drawbacks to using a binderless form of OPT. One of them is a lack of dimension stability [59]. Baskaran et al. [34] improved the properties of binderless materials by adding PHA using freeze-dried and pure samples. The addition of freeze-dried and pure PHA to the panels improved the overall properties of the samples. Additionally, compared to panels that were made without additives or controls, the panels that were made from steam-treated oil palm particles with varying amounts of PHA had improved physical and mechanical properties [31]. PHA appears to have some potential for enhancing the fundamental properties of experimental oil palm particleboard panels without adhesives. Baskaran et al. [30] optimized the press temperature and time for binderless particleboard that was manufactured from oil palm trunk biomass using a rotatable central composite design of response surface methodology. The optimization process was completed with the aid of the response surface method and software. Compared to the Japanese Industrial Standard (JIS), the optimal solutions that were proposed in this study were satisfactory.

Other techniques for enhancing the properties of binderless particleboard include being treated with hot water/sodium hydroxide (NaOH), steamed, or heated [26,28,29,32]. The MOE and MOR of samples that were prepared from raw material that was treated with hot water were greater than those of control panels that were prepared from NaOH-treated particles [32]. Saari et al. [26] reported that steaming improved the mechanical and physical properties of the boards. Steam exposure appears to be an inexpensive way to enhance the overall properties of such panels. The two-step post-heat treatment in OPT particles can enhance the dimensional stability and biological durability of the particleboard. Regrettably, the treatment significantly reduced the mechanical strength of the particleboard [29]. According to Komariah et al. [28], particleboard containing 10% ADP exhibited the highest MOR (8.9 MPa), IB (0.53 MPa), and thickness swelling (TS) values (5.9%). Additionally, particleboards containing 10% ADP demonstrated excellent resistance to water.

According to Yusof et al. [25] adding PVA, citric acid, and calcium carbonate to the composition of particleboards may also affect their fire resistance. Additionally, the addition of wood particles may improve the properties of binderless particleboard from OPT. The study was successfully developed by the addition of *Acacia mangium* wood which increased the fractal dimension of the particleboard, resulting in a decrease in hydrophilicity. The reduced percentage TS and WA properties of the binderless particleboard corroborate these findings [39,40].

### 7.3. Cement Board

OPT could be used as a natural fiber-based filler to manufacture gypsum composites with improved fire-retardant properties. The initial and final setting times of the OPT gypsum composite were significantly longer than the rice husk (RH) one. Compared to the RH gypsum composite, the OPT gypsum composite demonstrated superior mechanical properties [27]. Additionally, Phutthimethakul et al. [56] investigated the use of flue-gas desulfurization (FGD) gypsum, construction, and demolition waste (CDW), and OPT as part of the raw materials for the manufacture of concrete bricks. The compressive strength of the concrete brick specimens containing 5.5% gypsum without CDW and OPT was 45.18 MPa. In contrast, concrete brick specimens containing 5.5% FGD gypsum, 75% CDW, and 1% OPT had a compressive strength of 26.84 MPa.

## 8. Future Perspective

The future prospects of the utilization of OPT can be highlighted through the success of experimental works of engineered products such as the substitution of wood panel products, such as plywood, LVL, particleboard, fiberboard, and cement board as discussed above. In other words, OPT is potential to be a raw material for composite products which are usually made from wood. Challenges arose during the conversion of the OPT into veneer and lumber because of the monocotyledon characteristics which are rich in parenchyma and the irregular dimension of the fibers, thus after seasoning defects can be found remarkably. Board-based particles or fibers can be easily produced since transforming the OPT into both forms is now simply by applying a mechanical clearing system on the site plantation, where both types of dimensions are available afterwards. However, the determination of adhesive used still needed attention, whether its function as only a glue or at the same time performed as bonding agent, preservative chemical, and impregnation material. Even though both particleboard and fiberboard can be manufactured without adhesive, called binderless, care on hot-pressing stages should be carried out carefully. Wet processes or very high moisture within the particles was able to activate hydrogen bonding, thus the particles or fibers were interconnected each other. All this will be different to cement board for which the predominant component is cement or mineral. Cement board needs measurement of hydration temperature, which indicates the suitability of OPT with cement. A benefit would arise if both Indonesia and Malaysia, as the most oil palm producing countries, produced OSB, a structural panel product, since both countries have had no data on it.

## 9. Conclusions

Oil palm plantations have multiplied in Southeast Asia, particularly in Indonesia and Malaysia. Palm oil is used for the manufacture of multi-purpose products, ranging from edible to medicinal supplement and oleo-chemicals for household care products to biofuel, making it one of the most economically beneficial plants. After 25–30 years, the palms are felled and replaced due to declining oil production. OPT contributes significantly to waste because the trunks are then recycled or burned on the plantation. This increases insect and fungi populations, which poses difficulties for the new palm generation or contributes to the air pollution that is caused by the fire. If no effort is made to utilize oil palm biomass, it will become a problem for the environment. In fact, OPT is a sustainable non-wood source that is available in abundance at replanting operations, and would have seemed an alternative for raw materials for some of the products that are traditionally made from wood. Unfortunately, OPT is monocotyledon and consists of parenchyma and vascular bundles, which are the main obstacle in utilizing OPT as a solid. In contrast with wood, OPT has had higher moisture, dimensional instability, less durability, and poor machining. The oil palm industry must be prepared to capitalize on this situation and maximize the utilization of available OPT to convert waste to wealth, eliminate disposal issues, and generate value-added products by either raw material treatment or modification using resin. For the first, seasoning of OPT veneer using microwave, making hybrid plywood, and sandwiching or overlaying using rubberwood were examples for improving the dimensional stability of the final products. For the latter, impregnation using low molecular weight resin, treatment with several resins, and the addition of chemicals for binding activation were efforts for controlling the hygroscopicity and enhancing durability. Therefore, OPT would be utilized as alternative wood products, may be used in place of wood to manufacture wood-based panels because so far, many studies OPT could be engineered into the substitution of traditional wood products such as plywood, laminated veneer lumber (LVL), particleboard, and cement board or as the predominant matrix for end products of lumber, furniture, and lightweight construction. Unfortunately, the production of OSB in the two countries has not been carried out much. Therefore, this is the opportunity to develop the OSB industry that is made of vascular bundles in Indonesia and Malaysia.

## Figures and Tables

**Figure 1 polymers-14-01758-f001:**
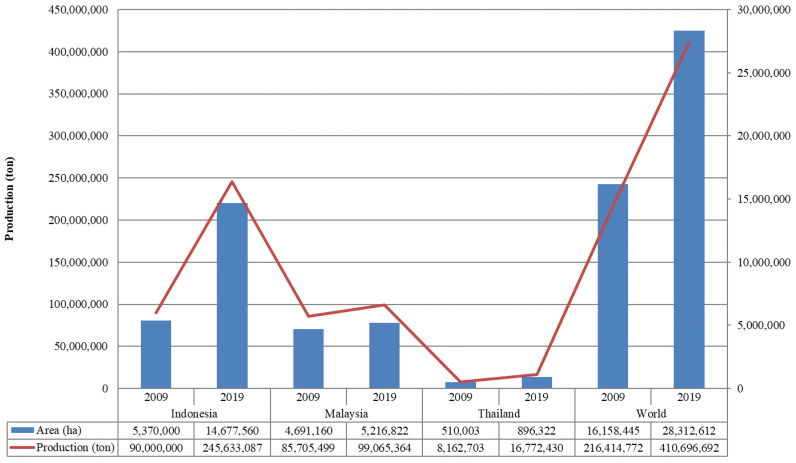
World area that was harvested and oil palm production within the predominant three countries in Southeast Asia during 2009–2019, Indonesia, Malaysia, and Thailand [12].

**Figure 2 polymers-14-01758-f002:**
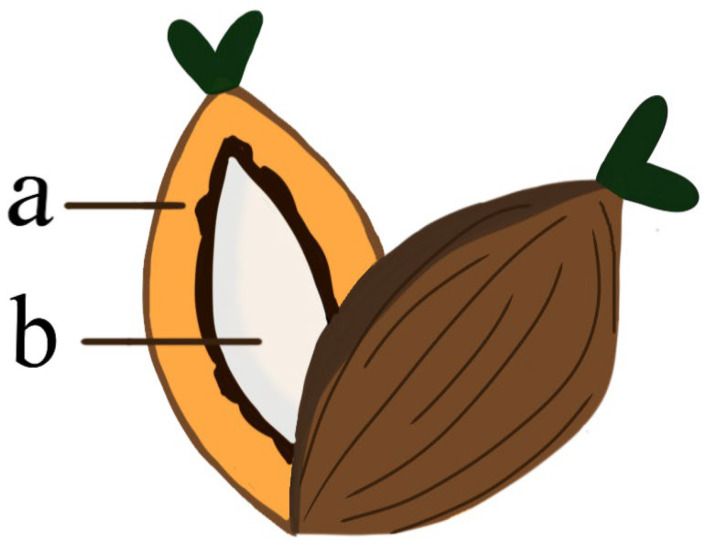
Cross-section of an oil palm fruit that is comprised of (**a**) an outer oily flesh or pericarp (consisted of exocarp, mesocarp, and endocarp) and (**b**) an oil-rich seed/nut or kernel (endosperm).

**Figure 3 polymers-14-01758-f003:**
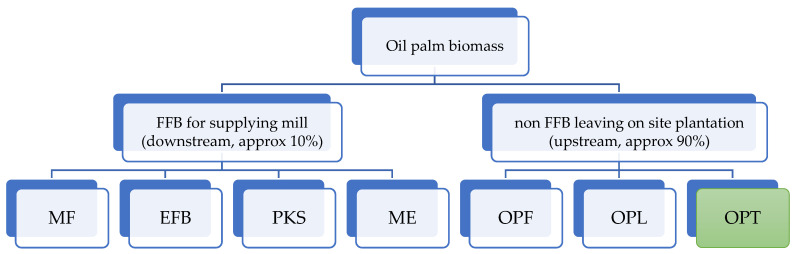
Biomass of oil palm that is generated both from downstream (fruit origin) and upstream (non-fruit). The focus of this study is OPT which shown in green color.

**Figure 4 polymers-14-01758-f004:**
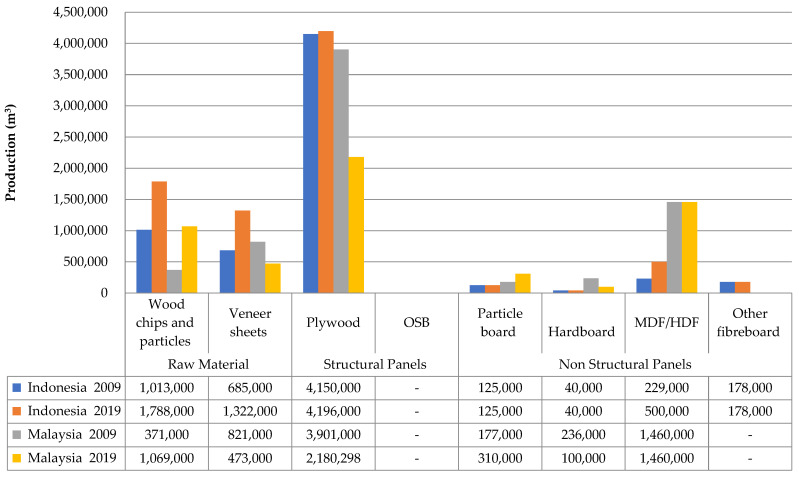
Data production of wood-based panels of Indonesia and Malaysia in a decade (2009–2019) consisting of raw material (wood chips and particles and veneer sheets), structural panels (plywood & OSB), and non-structural panels (particleboard and fiberboard) [12].

**Figure 5 polymers-14-01758-f005:**
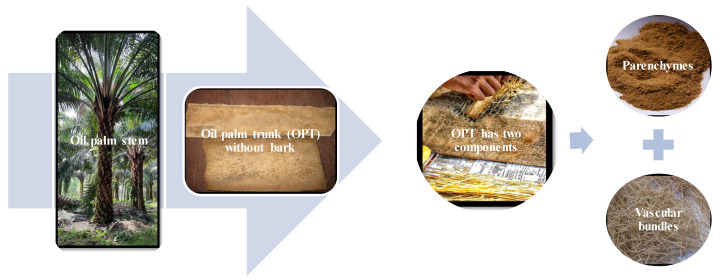
Two main parts of an oil palm trunk (OPT). For optimization, between the parenchyma and vascular bundles have been separated.

**Figure 6 polymers-14-01758-f006:**
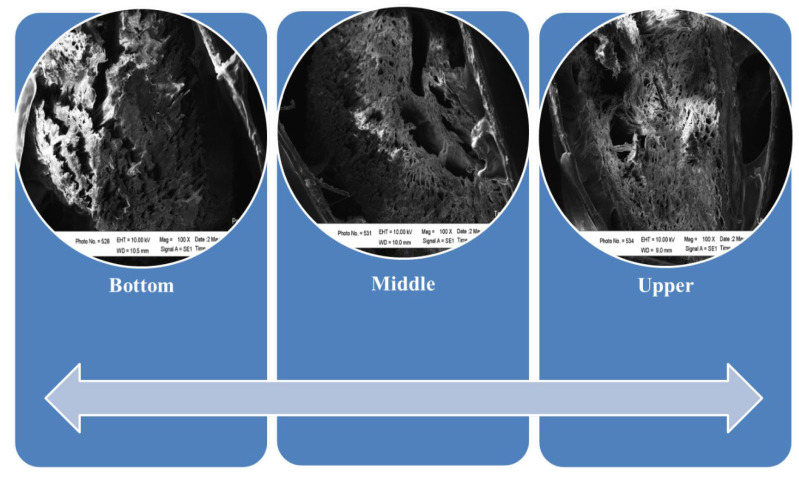
Vascular bundles of OPT showed no differences among the bottom, middle, and upper parts (100× magnifications).

**Table 1 polymers-14-01758-t001:** Results of the search engine sciencedirect.com uses the keywords: “oil palm trunk” OR “oil palm stem” OR “oil palm waste”.

Sciencedirect.com (31 Articles)			
No	Type of Panels	Year	Ref
1	Particleboard made of OPT bonded by poly(vinyl)alcohol/PVA	2020	[25]
2	Plywood made of OPT veneer without binder	2020	[26]
3	Cement (gypsum) board with OPT filler	2019	[27]
4	Binderless particleboard made of inner part of OPT	2019	[28]
5	OPT chips particleboard bonded by urea-formaldehyde (UF) resin	2018	[29]
6	OPT chips binderless particleboard with different thickness	2015	[30]
7	OPT fine particleboard with addition of polyhydroxylalkanoates (PHA)	2013	[31]
8	Particleboard made of treated OPT particles bonded by UF resin	2014	[32]
9	Laminated panels made of compressed OPT and poly(vinyl)acetate/PVAc	2013	[33]
10	OPT fine particleboard with addition of PHA	2012	[34]
11	Binderless OPT fine particleboard	2011	[35]
12	Three-layers laminated veneer lumber (LVL) made of OPT and UF resin	2013	[36]
13	Binderless particleboard made of OPT fine particles and strands	2010	[37]
14	Oil palm stem (OPS) plywood treated by phenol-formaldehyde (PF)	2011	[38]
15	Binderless particleboard surface made of OPT and acacia wood particles	2019	[39]
16	LVL made of OPT veneer with different types of formaldehyde resins	2009	[40]
17	Hybrid plywood made of OPT and oil palm empty fruit bunches (OPEFB)	2010	[41]
18	Plywood made of OPS pre-preg by PF	2013	[42]
19	Plywood, particleboard, and fiberboard made of OPT	2012	[43]
20	Properties of binderless particleboard made of OPT and acacia wood	2014	[44]
21	OPS plywood treated by low molecular weight PF	2012	[45]
22	Seasoning of veneer OPT	2017	[46]
23	Plywood made of OPT veneer treated with various concentration of PF	2013	[47]
24	Biocomposite made of oil palm biomass including OPT	2015	[48]
25	Binderless particleboard made of young and old OPT	2014	[49]
26	Binderless particleboard made of fine particles and vascular strands OPT	2014	[50]
27	Binderless fiberboard made of agricultural biomass including OPT	2019	[51]
28	Binderless particleboard made of oil palm biomass including OPT	2011	[52]
29	Laminated composites made of agricultural biomass including OPT	2017	[53]
30	OPT as sandwich panel core bonded by melamine-formaldehyde (MF)	2015	[54]
31	OPS veneer treated by low molecular weight PF	2011	[55]
mdpi.com (2 articles)			
No	Type of panels	Year	Reference
1	Concrete brick with cement, gypsum, gravel and OPT aggregate	2020	[56]
2	Hybrid plywood made of OPT veneer and OPEFB mat bonded by PF	2020	[57]

**Table 2 polymers-14-01758-t002:** Biomass residue of oil palm in two countries, Indonesia and Malaysia, that are derived from the fruit and non-fruit (million tonnes).

	Indonesia [22]	Malaysia [66]
**Fruit**		
Shell (PKS)	4.83	4.72
OPEFB	15.87	7.78
Fiber (MF)	8.97	8.18
POME	n.a	3.38
**Non-fruit**		
Frond (OPF)	43.05	23.39
Trunk (OPT/OPS)	13.94	3.74

Remarks: n.a is no data available.

**Table 3 polymers-14-01758-t003:** Several modifications of OPT for LVL and plywood panel.

Product	Adhesive/ Binder	Studies	Refs.
LVL	UF, PF, MF, and PRF	Variation of adhesive	[40]
Plywood	UF	UF impregnated OPT for core plywood	[36]
Plywood	PF	Treatment with LmwPF resin: effect of pressing pressure, the effect of resin content, and effect of hot-pressing time	[42,45,47]
Veneer	-	Effects of hot air and microwave drying	[46]
Composite-Plywood	UF and PF	Hybrid plywood from oil palm biomass	[57]
Composite-Plywood	MUF	Sandwich panel with OPT core overlaid with rubberwood veneer	[54]
Composite-Plywood	UF	Impregnated OPT with UF resin	[36]
Composite-Plywood	PF	Addition of OPA to hybrid plywood	[57]
Composite-Plywood	-	Binderless compressed veneer panel using response surface methodology	[26]

**Table 4 polymers-14-01758-t004:** Several studies and modifications oil palm trunk (OPT) for particleboard panel.

Product	Adhesive/ Binder	Studies	Ref.
Binderless Particleboard	-	Effect of particle geometry on binderless particleboard	[37]
Binderless Particleboard	-	Influence of press temperature on the properties of binderless particleboard	[35]
Binderless Particleboard	-	Determine the chemical component suitability for binderless particleboard	[59]
Binderless Particleboard	-	Effect of oil palm age on properties of binderless particleboard	[49]
Particleboard	PHA	Effect addition of PHA	[34]
Particleboard	PHA	Influence of steam treatment and addition of PHA.	[31]
Particleboard	-	Optimization of press temperature and time for binderless particleboard	[30]
Particleboard	UF	Effect of treated with hot water and sodium hydroxide (NaOH) on the properties of particleboard	[32]
Binderless Particleboard	-	Steam treated on binderless particleboard	[26]
Particleboard	UF	Effects of two-step post heat-treatment on the particleboard properties.	[29]
Binderless Particleboard	ADP	Addition of ammonium dihydrogen phosphate (ADP)	[28]
Binderless Particleboard	-	Addition *Acacia mangium* particle	[44]
Particleboard	PVA	Addition citric acid and calcium carbonate	[25]

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
