# Peer review of "Panel Products Made of Oil Palm Trunk: A Review of Potency, Environmental Aspect, and Comparison with Wood-Based Composites"

_polymers, 2022, doi:10.3390/polym14091758_

Round 1

Reviewer 1 Report

Manuscript No.: polymers-1648111-peer-review-v1

Title: Composite Panel Products Made of Oil Palm Trunk: A Review

Polymers

The article traces interesting review studies about “Composite Panel Products Made of Oil Palm Trunk”. The availability of literature adheres to the expected standards of your research community. After major corrections to the form and content of this version, the manuscript will be ready for publication. However, there are still unclear issues that need reformulation and/or re-organization.

Title

  • The title should be well defined and more comprehensive to provide the detaild of the review.

Abstract

  • The abstract should be more descriptive with details rather than just discussion.

Introduction

  • The introduction section is well written but some results from the studies must be add to increase interest of the readers.

Conclusion

  • A paragraph of future perspective should be added.

Conclusion

  • The conclusion section should be more comparison based that literature you have surveyed.

References

  • Normally a review article describes the literature review of the last five years and is better to replace old references with the latest or last five-year (2018-2021/2022) published literature.

Note: English editing is required and the manuscript must be proofread by the native English speaker to address language issues.

Author Response

Thank you for your effort to handling our manuscript ID polymers-1648111. Authors also acknowledged to Reviewer who made this quality of the manuscript improved. Here, author’s response to reviewer’s comments.

Reviewer 2 Report

General

The article summarizes in detail various works describing the properties and uses of palm trunks. It also summarizes the strengths and weaknesses of palm trunk remains after the end of their life. It highlights the possibilities of using palm trunk for the production of particleboard based on various studies.

Comments

Despite the merits of the article, in many cases it describes the basic facts that are clearly described in detail in many literatures. For example, a description of the composition of spruce (L448 - 454) without a direct comparison of palm and spruce tissue. On the other hand, concrete data for the use of palm trunk are missing. For example, the description of mechanical properties (L377-L395) lacks tissue strength, which is important for use.

It is suggested to the authors to give more concrete information in the review, preferably in graphical form, showing certain relationships, so that the reader gets a better insight into the different researches studied and does not have to read additionally the whole literature for rough information.

Author Response

Thank you very much for taking the time to look over our manuscript ID polymers-1648111. Authors also expressed gratitude to the Reviewer for helping to improve the manuscript's quality. The author's answer to the reviewer's comments can be found here.

Reviewer 3 Report

The paper deals with a very actual topic. It is well written and has a good and logical structure.

I suggest the paper for publication

Author Response

Thank you for taking the time to read through our paper ID polymers-1648111. The authors additionally thanked the Reviewer for the very positive comments.

Round 2

Reviewer 1 Report

The said comments has been addressed successfully and the manuscript can be accepted after fulfilling protocols and SOPs of Polymers. Some minor English issues can be settled during proofreading.